# Decentralized Learning for Overparameterized Problems: A Multi-Agent Kernel Approximation Approach

**Prashant Khanduri**[†‡]**, Haibo Yang**[‡]**, Mingyi Hong**[†]**, Jia Liu**[‡]**, Hoi-To Wai**[°]**, Sijia Liu**[◇∗]
[†]University of Minnesota, [‡]The Ohio State University, [°]CUHK, [◇]Michigan State University,
[∗]MIT-IBM Watson AI Lab, IBM Research
`khand095@umn.edu, yang.5952@osu.edu, mhong@umn.edu`
`liu@ece.osu.edu, htwai@se.cuhk.edu.hk, liusijia5@msu.edu`

## Abstract

This work develops a novel framework for communication-efficient distributed learning where the models to be learnt are overparameterized. We focus on a class of kernel learning problems (which includes the popular neural tangent kernel (NTK) learning as a special case) and propose a novel *multi-agent kernel approximation* technique that allows the agents to distributedly estimate the full kernel function, and subsequently perform distributed learning, without directly exchanging any local data or parameters. The proposed framework is a significant departure from the classical consensus-based approaches, because the agents do not exchange problem parameters, and consensus is not required. We analyze the optimization and the generalization performance of the proposed framework for the $\ell_2$ loss. We show that with $M$ agents and $N$ total samples, when certain generalized inner-product (GIP) kernels (resp. the random features (RF) kernel) are used, each agent needs to communicate $\mathcal{O}(N^2/M)$ bits (resp. $\mathcal{O}(N\sqrt{N}/M)$ real values) to achieve minimax optimal generalization performance. Further, we show that the proposed algorithms can significantly reduce the communication complexity compared with state-of-the-art algorithms, for distributedly training models to fit UCI benchmarking datasets. Moreover, each agent needs to share about $200N/M$ bits to closely match the performance of the centralized algorithms, and these numbers are independent of parameter and feature dimension.

## 1 Introduction

Recently, decentralized optimization has become a mainstay of the optimization research. In decentralized optimization, multiple local agents hold small to moderately sized private datasets, and collaborate by iteratively solving their local problems while sharing some information with other agents. Most of the existing decentralized learning algorithms are deeply rooted in classical consensus-based approaches (Tsitsiklis, 1984), where the agents repetitively share the local parameters with each other to reach an optimal *consensual* solution. However, the recent trend of using learning models in the *overparameterized* regime with very high-dimensional parameters (He et al., 2016; Vaswani et al., 2017; Fedus et al., 2021) poses a significant challenge to such parameter sharing approaches, mainly because sharing model parameters iteratively becomes excessively expensive as the parameter dimension grows. If the size of local data is much smaller than that of the parameters, perhaps a more efficient way is to directly share the local data. However, this approach raises privacy concerns, and it is rarely used in practice. Therefore, a fundamental question of decentralized learning in the overparameterized regime is:

> **(Q)** For overparameterized learning problems, how to design decentralized algorithms that achieve *the best* optimization/generalization performance by exchanging *minimum* amount of information?

We partially answer **(Q)** in the context of distributed kernel learning (Vert et al., 2004). We depart from the popular consensus-based algorithms and propose an optimization framework that does not require the local agents to share model parameters or raw data. We focus on kernel learning because: (i) kernel methods provide an elegant way to model non-linear learning problems with complex data

Table 1: Comparison of the total communication required per node by different algorithms for non-overparameterized (NOP) and overparameterized (OP) regimes. Please see Appendix B for a detailed discussion of the algorithms. Here $N$ is entire sample size, UB on $M$ denotes the upper bound on the number of nodes, $M$, $d$ is the data dimension, $\beta \geq 2$ is a constant, and $T$ denotes the total communication (iterations) rounds utilized by the distributed algorithms.

| Algorithm | Kernel | UB on $M$ | Communication (Real Values) | |
| | | | NOP | OP |
| --- | --- | --- | --- | --- |
| DKRR-CM (Lin et al., 2020) | Any | $\mathcal{O}\big(N^{\frac{T+1}{2(T+2)}}\big)$ | $\mathcal{O}(dTN)$ | $\mathcal{O}(dTN)$ |
| DKRR-RF-CM (Liu et al., 2021) | RF | $\mathcal{O}\big(N^{\frac{T+1}{2(T+2)}}\big)$ | $\mathcal{O}(T\sqrt{N})$ | $\mathcal{O}(TN^{\beta})$ |
| Decentralized-RF (Richards et al., 2020) | RF | $\mathcal{O}(N^{\frac{1}{3}})$ | $\mathcal{O}(T\sqrt{N})$ | $\mathcal{O}(TN^{\beta})$ |
| DKLA/COKE (Xu et al., 2020) | RF | Any $M$ | $\mathcal{O}(T\sqrt{N})$ | $\mathcal{O}(TN^{\beta})$ |
| Algorithm 2 (this work) | RF | Any $M$ | $\mathcal{O}\big(\frac{N\sqrt{N}}{M}\big)$ | $\mathcal{O}\big(\frac{N^{1+\beta}}{M}\big)$ |
| | GIP | | $\mathcal{O}\big(\frac{N^2}{M}\big)$ | $\mathcal{O}\big(\frac{N^2}{M}\big)$ |

dependencies as simple linear problems (Vert et al., 2004; Hofmann et al., 2008), and (ii) kernel-based methods can be used to capture the behavior of a fully-trained deep network with large width (Jacot et al., 2018; Arora et al., 2019; 2020).

Distributed implementation of kernel learning problems is challenging. Current state-of-the-art algorithms for kernel learning either rely on sharing raw data among agents and/or imposing restrictions on the number of agents (Zhang et al., 2015; Lin et al., 2017; Koppel et al., 2018; Lin et al., 2020; Hu et al., 2020; Pradhan et al., 2021; Predd et al., 2006). Some recent approaches rely on specific random feature (RF) kernels to alleviate some of the above problems. These algorithms reformulate the (approximate) problem in the parameter domain and solve it by iteratively sharing the (potentially high-dimensional) parameters (Bouboulis et al., 2017; Richards et al., 2020; Xu et al., 2020; Liu et al., 2021). These algorithms suffer from excessive communication overhead, especially in the overparameterized regime where the number of parameters is larger than the data size $N$. For example, implementing the neural tangent kernel (NTK) with RF kernel requires at least $\mathcal{O}(N^{\beta})$, $\beta > 2$, random features (parameter dimension) using ReLU activation (Arora et al., 2019; Han et al., 2021)[1]. For such problems, in this work, we propose a novel algorithmic framework for decentralized kernel learning. Below, we list the major contributions of our work.

**[GIP Kernel for Distributed Approximation]** We define a new class of kernels suitable for distributed implementation, Generalized inner-product (GIP) kernel, that is fully characterized by the angle between a pair of feature vectors and their respective norms. Many kernels of practical importance including the NTK can be represented as GIP kernel. Further, we propose a *multi-agent kernel approximation* method for estimating the GIP and the popular RF kernels at individual agents.

**[One-shot and Iterative Scheme]** Based on the proposed kernel approximation, we develop two optimization algorithms, where the first one only needs *one-shot* information exchange, but requires sharing data labels among the agents; the second one needs *iterative* information exchange, but does not need to share the data labels. A key feature of these algorithms is that neither the raw data features nor the (high-dimensional) parameters are exchanged among agents.

**[Performance of the Approximation Framework]** We analyze the optimization and the generalization performance of the proposed approximation algorithms for $\ell_2$ loss. We show that GIP kernel requires communicating $\mathcal{O}(N^2/M)$ bits and the RF kernel requires communicating $\mathcal{O}(N\sqrt{N}/M)$ real values per agent to achieve minimax optimal generalization performance. Importantly, the required communication is independent of the function class and the optimization algorithm. We validate the performance of our approximation algorithms on UCI benchmarking datasets.

In Table 1, we compare the communication requirements of the proposed approach to popular distributed kernel learning algorithms. Specifically, DKRR-CM (Lin et al., 2020) relies on sharing data and is therefore not preferred in practical settings. For the RF kernel, the proposed algorithm outperforms other algorithms in both non-overparameterized and the overparameterized regimes when $T > N/M$. In the overparameterized regime, the GIP kernel is more communication efficient compared to other algorithms. Finally, note that since our analysis is developed using the multi-agent-kernel-approximation, it does not impose any upper bound on the number of agents in the network.

---

[1]To achieve approximation error $\epsilon = \mathcal{O}(1/\sqrt{N})$.

**Notations:** We use $\mathbb{R}$, $\mathbb{R}^d$, and $\mathbb{R}^{n \times m}$ to denote the sets of real numbers, $d$-dimensional Euclidean space, and real matrices of size $n \times m$, respectively. We use $\mathbb{N}$ to denote the set of natural numbers. $\mathcal{N}(0, \Sigma)$ is multivariate normal distribution with zero mean and covariance $\Sigma$. Uniform distribution with support $[a, b]$ is denoted by $\mathcal{U}[a, b]$. $\langle a, b \rangle$ (resp. $\langle a, b \rangle_{\mathcal{H}}$) denotes the inner-product in Euclidean space (resp. Hilbert space $\mathcal{H}$). The inner-product defines the usual norms in corresponding spaces. Norm $\|A\|$ of matrix $A$ denotes the operator norm induced by $\ell_2$ vector norm. We denote by $[a]_i$ or $[a]^{(i)}$ the $i^{\text{th}}$ element of a vector $a$. $[A \cdot a]_j^{(i)}$ denotes the $(i \cdot j)^{\text{th}}$ element of vector $A \cdot a$. Moreover, $A^{(:,i)}$ is the $i^{\text{th}}$ column of $A$ and $[A]_{mk}$ is the element corresponding to $m^{\text{th}}$ row and $k^{\text{th}}$ column. Notation $m \in [M]$ denotes $m \in \{1, .., M\}$. Finally, $\mathbb{1}[E]$ is the indicator function of event $E$.

## 2 PROBLEM STATEMENT

Given a probability distribution $\pi(x, y)$ over $\mathcal{X} \times \mathbb{R}$, we want to minimize the population loss

$$\mathcal{L}(f) = \mathbb{E}_{x,y \sim \pi(x,y)}[\ell(f(x), y)], \tag{1}$$

where $x \in \mathcal{X} \subset \mathbb{R}^d$ and $y \in \mathbb{R}$ denote the features and the labels, respectively. Here, $f : \mathcal{X} \to \mathbb{R}$ is an estimate of the true label $y$. We consider a distributed system of $M$ agents, with each agent $m \in [M]$ having access to a locally available independently and identically distributed (i.i.d) dataset $\mathcal{N}_m = \{x_m^{(i)}, y_m^{(i)}\}_{i=1}^n$ with[2] $(x_m^{(i)}, y_m^{(i)}) \sim \pi(x, y)$. The total number of samples is $N = nM$. The goal of kernel learning with kernel function, $k(\cdot, \cdot) : \mathcal{X} \times \mathcal{X} \to \mathbb{R}$, is to find a function $f \in \mathcal{H}$ (where $\mathcal{H}$ is the reproducing kernel Hilbert space (RKHS) associated with $k$ (Vert et al., 2004)) that minimizes (1). We aim to solve the following (decentralized) empirical risk minimization problem

$$\min_{f \in \mathcal{H}} \left\{ \hat{\mathcal{R}}(f) = \hat{\mathcal{L}}(f) + \frac{\lambda}{2}\|f\|_{\mathcal{H}}^2 = \frac{1}{M}\sum_{m=1}^{M} \hat{\mathcal{L}}_m(f) + \frac{\lambda}{2}\|f\|_{\mathcal{H}}^2 \right\}, \tag{2}$$

where $\lambda > 0$ is the regularization parameter and $\hat{\mathcal{L}}_m(f) = \frac{1}{n}\sum_{i \in \mathcal{N}_m} \ell(f(x_m^{(i)}), y_m^{(i)})$ is the local loss at each $m \in [M]$. Problem (2) can be reformulated using the Representer theorem (Schölkopf et al., 2002) with $\hat{\mathcal{L}}_m(\alpha) = \frac{1}{n}\sum_{i \in \mathcal{N}_m} \ell([\mathbf{K}\alpha]_m^{(i)}, y_m^{(i)}), \forall m \in [M]$, as

$$\min_{\alpha \in \mathbb{R}^N} \left\{ \hat{\mathcal{R}}(\alpha) = \hat{\mathcal{L}}(\alpha) + \frac{\lambda}{2}\|\alpha\|_{\mathbf{K}}^2 = \frac{1}{M}\sum_{m=1}^{M} \hat{\mathcal{L}}_m(\alpha) + \frac{\lambda}{2}\|\alpha\|_{\mathbf{K}}^2 \right\}, \tag{3}$$

where $\mathbf{K} \in \mathbb{R}^{N \times N}$ is the kernel matrix with elements $k(x_m^{(i)}, x_{\bar{m}}^{(j)})$, $\forall m, \bar{m} \in [M]$, $\forall i \in \mathcal{N}_m$ and $\forall j \in \mathcal{N}_{\bar{m}}$. The supervised (centralized) learning problem (3) is a classical problem in statistical learning (Caponnetto & De Vito, 2007) and has been popularized recently due to connections with overparameterized neural network training (Jacot et al., 2018; Arora et al., 2019). An alternate way to solve problem (2) (and (3)) is by parameterizing $f$ in (2) by $\theta \in \mathbb{R}^D$ as $f_D(x; \theta) = \langle \theta, \phi_D(x) \rangle$ where $\phi_D : \mathcal{X} \to \mathbb{R}^D$ is a finite dimensional feature map. Here, $\phi_D(\cdot)$ is designed to approximate $k(\cdot, \cdot)$ with $k_D(x, x') = \langle \phi_D(x), \phi_D(x') \rangle$ (Rahimi & Recht, 2008). Using this approximation, problem (2) (and (3)) can be written in the parameter domain with $\hat{\mathcal{L}}_{m,D}(\theta) = \frac{1}{n}\sum_{i \in \mathcal{N}_m} \ell(\langle \theta, \phi_D(x_m^{(i)}) \rangle, y_m^{(i)}), \forall m \in [M]$, as

$$\min_{\theta \in \mathbb{R}^D} \left\{ \hat{\mathcal{R}}_D(\theta) = \hat{\mathcal{L}}_D(\theta) + \frac{\lambda}{2}\|\theta\|^2 = \frac{1}{M}\sum_{m=1}^{M} \hat{\mathcal{L}}_{m,D}(\theta) + \frac{\lambda}{2}\|\theta\|^2 \right\}. \tag{4}$$

Note that (4) is a $D$-dimensional problem, whereas (3) is an $N$-dimensional problem. Since (4) is in the standard finite-sum form, it can be solved using the standard *parameter sharing* decentralized optimization algorithms (e.g., DGD (Richards et al., 2020) or ADMM (Xu et al., 2020) ), which share $D$-dimensional vectors iteratively. However, when (4) is *overparameterized* with very large $D$ (e.g., $D = \mathcal{O}(N^{\beta})$ with $\beta \geq 2$ for the NTK), such parameter sharing approaches are no longer feasible because of the increased communication complexity. An intuitive solution to avoid sharing these high-dimensional parameters is to directly solve (3). However, it is by no means clear if and how one can efficiently solve (3) in a *decentralized manner*. The key challenge is that, unlike the

---

[2]The techniques presented in this work can be easily extended to unbalanced datasets, i.e., when each agent has a dataset of different size.

conventional decentralized learning problems, here each loss term $\ell([\mathbf{K}\alpha]_m^{(i)}, y_m^{(i)})$ is not separable over the agents. Instead, each agent $m$'s local problem is dependent on $k(x_m^{(i)}, x_{\bar{m}}^{(j)})$ with $m \neq \bar{m}$. Importantly, without directly transmitting the data itself (as has been done in Predd et al. (2006); Koppel et al. (2018); Lin et al. (2020)), it is not clear how one can obtain the required $(m \cdot i)^{\text{th}}$ element of $\mathbf{K}\alpha$. Therefore, to develop algorithms that avoid sharing high-dimensional parameters by directly (approximately) solving (3), it is important to identify kernels that are suitable for decentralized implementation and propose efficient algorithms for learning with such kernels.

## 3 THE PROPOSED ALGORITHMS

In this section, we define a general class of kernels referred to as the *generalized inner product (GIP) kernels* that are suitable for decentralized overparameterized learning. By focusing on GIP kernels, we aim to understand the best possible decentralized optimization/generalization performance that can be achieved for solving (3). Surprisingly, one of our proposed algorithm only shares $\mathcal{O}(nN) = \mathcal{O}(N^2/M)$ bits of information per node, while achieving the minimax optimal generalization performance. Such an algorithm only requires one round of communication, where the messages transmitted are independent of the actual parameter dimension (i.e., $D$ in problem (4)); further, there is no requirement for achieving consensus among the agents. The proposed algorithm represents a significant departure from the *classical* consensus-based decentralized learning algorithms. We first define a class of kernels that we will focus on in this work.

**Definition 3.1. [Generalized inner-product (GIP) kernel]** We define a GIP kernel as:

$$k(x, x') = g(\psi(x, x'), \|x\|, \|x'\|), \tag{5}$$

where $\psi(x, x') = \arccos(x^T x'/(\|x\|\|x'\|)) \in [0, \pi]$ denotes the angle between the feature vectors $x$ and $x'$; and $g(\cdot, \|x\|, \|x'\|)$ is assumed to be Lipschitz continuous (cf. Assumption 2). □

*Remark* 1. Note that the GIP kernel is a generalization of the inner-product kernels (Schölkopf et al., 2002), i.e., kernels of the form $k(x, x') = k(\langle x, x' \rangle)$. Clearly, $k(\langle x, x' \rangle)$ can be represented as $k(\langle x, x' \rangle) = g(\psi(x, x'), \|x\|, \|x'\|)$ for some function $g(\cdot)$. Moreover, many kernels of practical interest can be represented as GIP kernels, some examples include NTK (Jacot et al., 2018; Chizat et al., 2019; Arora et al., 2019), arccosine (Cho & Saul, 2009), polynimal, Gaussian, Laplacian, sigmoid, and inner-product kernels (Schölkopf et al., 2002).

The main reason we focus on the GIP kernels for decentralized implementation is that, this class of kernels can be fully specified at each agent if the norms of all the feature vectors and the pairwise angles between them are known at each agent. For example, consider an NTK of a single hidden-layer ReLU neural network: $k(x, x') = x^T x'(\pi - \psi(x, x'))/2\pi$ (Chizat et al., 2019). This kernel can be fully learned with just the knowledge of norms and the pairwise angles of the feature vectors. For many applications of interest (Bietti & Mairal, 2019; Geifman et al., 2020; Pedregosa et al., 2011), normalized feature vectors are used, and for such problems, the GIP kernel at each agent can be computed only by using the knowledge of the pairwise angles between the feature vectors. We show in Sec. 3.1 that such kernels can be efficiently estimated by each agent while sharing only a few bits of information. Importantly, the communication requirement for such a kernel estimation procedure is independent of the problem's parameter dimension (i.e., $D$ in (4)), making them suitable for decentralized learning in overparameterized regime. Next, we define the RF kernel.

**Definition 3.2. [Random features (RF) kernel]** RF kernel is defined as (Rahimi & Recht, 2008; Rudi & Rosasco, 2017; Li et al., 2019):

$$k(x, x') = \int_{\omega \in \Omega} \bar{\zeta}(x, \omega) \cdot \bar{\zeta}(x', \omega) dq(\omega) \tag{6}$$

with $(\Omega, q)$ being the probability space and $\bar{\zeta} : \mathcal{X} \times \Omega \to \mathbb{R}$. □

*Remark* 2. The RF kernel can be approximated as: $k(\cdot, \cdot) \approx k_P(x, x') = \langle \phi_P(x), \phi_P(x') \rangle$, with $\phi_P(x) = \frac{1}{\sqrt{P}}[\bar{\zeta}(x, \omega_1), \ldots, \bar{\zeta}(x, \omega_P)]^T \in \mathbb{R}^P$ and $\{\omega_i\}_{i=1}^P$ drawn i.i.d. from distribution $q(\omega)$. A popular example of the RF kernels is the shift-invariant kernels, i.e., kernels of the form $k(x, x') = k(x - x')$ (Rahimi & Recht, 2008). The RF kernels generalize the random Fourier features construction (Rudin, 2017) for shift-invariant kernels to general kernels. Besides the shift-invariant kernels, important examples of the RF kernels include the inner-product (Kar & Karnick, 2012), and the homogeneous additive kernels (Vedaldi & Zisserman, 2012).

---

**Algorithm 1** Approximation: Local Kernel Estimation

---

1: **Initialize:** Distribution $p(\omega)$ over space $(\Omega, p)$ and mapping $\zeta : \mathcal{X} \times \Omega \to \mathbb{R}$ (see Section 3.1)
2: **for** $m \in [M]$ **do**
3:      Draw $P$ i.i.d. random variables $\omega_i \in \mathbb{R}^d$ with $\omega_i \sim p(\omega)$ for $i = 1, \ldots, P$
4:      Compute $\zeta(x_m^{(i)}, \omega_j) \ \forall i \in \mathcal{N}_m$ and $j \in [P]$
5:      Construct the matrix $A_m \in \mathbb{R}^{P \times n}$ with the $(i, j)$th element as $\zeta(x_m^{(i)}, \omega_j)$
6:      Communicate $A_m$ to every other agent and receive $A_{\bar{m}}$ with $\bar{m} \neq m$ from other agents
7:      If GIP is used, and data is not normalized, then communicate $\|x_m^{(i)}\|, \ \forall i \in \mathcal{N}_m$
8:      Estimate the kernel matrix $\mathbf{K}_P$ locally using (7) for the GIP and (9) for the RF kernel
9: **end for**

---

Next, we propose a multi-agent approximation algorithm to effectively learn the GIP and the RF kernels at each agent, as well as the optimization algorithms to efficiently solve the learning problem. Our proposed algorithms will follow an *approximation – optimization* strategy, where the agents first exchange some information so that they can locally *approximate* the full kernel matrix $\mathbf{K}$; then they can independently *optimize* the resulting approximated local problems. Below we list a number of key design issues arising from implementing such an approximation – optimization strategy:

**[Kernel approximation]** How to accurately approximate the kernel $\mathbf{K}$, locally at each agent? For example, for the GIP kernels, how to accurately estimate the angles $\psi(x_m^{(i)}, x_{\bar{m}}^{(j)})$ at a given agent $m$, where $j \in \mathcal{N}_{\bar{m}}$ and $\bar{m} \neq m$? This is challenging, especially when raw data sharing is not allowed.

**[Effective exchange of local information]** How shall we design appropriate messages to be exchanged among the agents? The type of messages that gets exchanged will be dependent on the underlying kernel approximation schemes. Therefore, it is critical that proposed approximation methods are able to utilize as little information from other agents as possible.

**[Iterative or one-shot scheme]** It is not clear if such an *approximation – optimization* scheme should be *one-shot* or *iterative* – that is, whether it is favourable that the agents *iteratively* share information and perform local optimization (similar to classical consensus-based algorithms), or they should do it just once. Again, this will be dependent on the underlying information sharing schemes.

Next, we will formally introduce the proposed algorithms. Our presentation follows the *approximation – optimization* strategy outlined above. We first discuss the proposed decentralized kernel approximation algorithm, followed by two different ways of performing decentralized optimization.

## 3.1 MULTI-AGENT KERNEL APPROXIMATION

The kernel $\mathbf{K}$ is approximated locally at each agent using Algorithm 1. Note that in Step 3, each agent randomly samples $\{\omega_i\}_{i=1}^{P}$ from distribution $p(\omega)$. This can be easily established via random seed sharing as in Xu et al. (2020); Richards et al. (2020). In Step 6, each agent shares a locally constructed matrix $A_m$ of size $P \times n$, whose elements $\zeta(x_m^{(i)}, \omega_i)$ will be defined shortly. The choices of $p(\omega)$ and $\zeta(\cdot, \cdot)$ in Step 1 depend on the choice of kernel. Specifically, we have:

**[Approximation for GIP kernel]** For the GIP kernel, we first assume that the feature vectors are normalized (Pedregosa et al., 2011). We then choose $p(\omega)$ to be any circularly symmetric distribution, for simplicity we choose $p(\omega)$ as $\mathcal{N}(0, I_d)$. Moreover, we use $\zeta(x, \omega) = \mathbb{1}[\omega^T x \geq 0]$ such that $A_m$ is a binary matrix with entries $\{0, 1\}$. Note that such matrices are easy to communicate. Next, we approximate the kernel $\mathbf{K}$ with $\mathbf{K}_P$ as

$$k(x_m^{(i)}, x_{\bar{m}}^{(j)}) \approx k_P(x_m^{(i)}, x_{\bar{m}}^{(j)}) = g(\psi_P(x_m^{(i)}, x_{\bar{m}}^{(j)}), \|x_m^{(i)}\|, \|x_{\bar{m}}^{(j)}\|), \tag{7}$$

where $k(x_m^{(i)}, x_{\bar{m}}^{(j)})$ and $k_P(x_m^{(i)}, x_{\bar{m}}^{(j)}) \ \forall i \in \mathcal{N}_m, \ \forall m \in [M]$ and $\forall j \in \mathcal{N}_{\bar{m}}$ and $\forall \bar{m} \in [M]$ are the individual elements of $\mathbf{K}$ and $\mathbf{K}_P$, resp., and $\psi_P(x_m^{(i)}, x_{\bar{m}}^{(j)})$ is an approximation of the angle $\psi(x_m^{(i)}, x_{\bar{m}}^{(j)})$ evaluated using $A_m, A_{\bar{m}}$ as

$$\psi(x_m^{(i)}, x_{\bar{m}}^{(j)}) \approx \psi_P(x_m^{(i)}, x_{\bar{m}}^{(j)}) = \left| \pi - 2\pi[A_m^{(:,i)}]^T [A_{\bar{m}}^{(:,j)}]/P \right|, \tag{8}$$

---

**Algorithm 2** Optimization: One-Shot Communication for Kernel Learning

---

1: **Initialize:** $\alpha_m^1 \in \mathbb{R}^N$, step-sizes $\{\eta_m^t\}_{t=1}^{T_m}$ at each agent $m \in [M]$
2: **for** $m \in [M]$ **do**
3:         Using Algorithm 1 construct $\mathbf{K}_P$
4:         Communicate $\bar{y}_m = [y_m^{(1)}, \ldots, y_m^{(n)}]^T \in \mathbb{R}^n$
5:         Using $\mathbf{K}_P$ and $\bar{y}_m$ construct $\hat{\mathcal{L}}_P(\alpha)$ (cf. (10)) locally using $\hat{\mathcal{L}}_{m,P}(\alpha)$
6:         **Option I:** Solve (10) exactly at each agent
7:         **Option II:** Solve (10) inexactly using GD at each agent
8:             **for** $t = 1$ to $T_m$
9:                 `GD Update:` $\alpha_m^{t+1} = \alpha_m^t - \eta_m^t \nabla \hat{\mathcal{R}}_P(\alpha_m^t)$
10:             **end for**
11: **end for**
12: **Return:** $\alpha_m^{T+1}$ for all $m \in [M]$

---

This implies that $\mathbf{K}$ can be approximated for the GIP kernel by communicating only $nP$ bits of information per agent. Note that in the general case if the feature vectors are *not* normalized, then (7) can be evaluated by communicating additional $n$ real values of the norms of the feature vectors by each agent; see Step 7 in Algorithm 1.

**[Approximation for RF kernel]** For the RF kernel, we choose $\zeta(\cdot, \cdot) = \bar{\zeta}(\cdot, \cdot)$ and $p(\omega) = q(\omega)$ as defined in (6) and approximate $\mathbf{K}$ with $\mathbf{K}_P$ as

$$k(x_m^{(i)}, x_{\bar{m}}^{(j)}) \approx k_P(x_m^{(i)}, x_{\bar{m}}^{(j)}) = \langle \phi_P(x_m^{(i)}), \phi_P(x_{\bar{m}}^{(j)}) \rangle, \tag{9}$$

where $k(x_m^{(i)}, x_{\bar{m}}^{(j)})$ and $k_P(x_m^{(i)}, x_{\bar{m}}^{(j)})$ are elements of $\mathbf{K}$ and $\mathbf{K}_P$, resp., $\phi_P(x_m^{(i)}) = 1/\sqrt{P}[A_m^{(:,i)}]$ and $\phi_P(x_{\bar{m}}^{(j)}) = 1/\sqrt{P}[A_{\bar{m}}^{(:,j)}]$. Note that $\mathbf{K}$ can be approximated for the RF kernel by sharing only $nP$ real values per agent. Further, the distribution $q(\omega)$ and the mapping $\bar{\zeta}(\cdot, \cdot)$ depend on the type of RF kernel used. For example, for shift-invariant kernels with random Fourier features, we can choose $\bar{\zeta}(x, \omega) = \sqrt{2} \cos(\omega^T x + b)$ with $\omega \sim q(\omega)$ and $b \sim \mathcal{U}[0, 2\pi]$ (Rahimi & Recht, 2008).

Now that using Algorithm 1 we have approximated the kernel matrix at all the agents, we are ready to solve (3) approximately.

## 3.2 THE DECENTRALIZED OPTIMIZATION STEP

The approximated kernel regression problem (3) with $\mathbf{K}_P$ obtained using Algorithm 1, and local loss $\hat{\mathcal{L}}_{m,P}(\alpha) := \frac{1}{n} \sum_{i \in \mathcal{N}_m} \ell([\mathbf{K}_P \alpha]_m^{(i)}, y_m^{(i)})$ is

$$\min_{\alpha \in \mathbb{R}^N} \left\{ \hat{\mathcal{R}}_P(\alpha) = \hat{\mathcal{L}}_P(\alpha) + \frac{\lambda}{2} \|\alpha\|_{\mathbf{K}_P}^2 = \frac{1}{M} \sum_{m=1}^M \hat{\mathcal{L}}_{m,P}(\alpha) + \frac{\lambda}{2} \|\alpha\|_{\mathbf{K}_P}^2 \right\}. \tag{10}$$

*Remark* 3. For the approximate problem (10), we would want $\mathbf{K}_P$ constructed using the multi-agent kernel approximation approach to be positive semi-definite (PSD), i.e., the kernel function $k_P(\cdot, \cdot)$ is a positive definite (PD) kernel. From the definition of the approximate RF kernel (9), it is easy to verify that it is PD. However, it is not clear if the approximated GIP kernel is PD. Certainly, for the GIP kernel we expect that as $P \to \infty$ we have $\mathbf{K}_P \to \mathbf{K}$, i.e., asymptotically $\mathbf{K}_P$ is PSD, since $\mathbf{K}$ is PSD. In the Appendix, we introduce a sufficient condition (Assumption 6) that ensures $\mathbf{K}_P$ to be PSD for the GIP kernel. In the following, for simplicity we assume $\mathbf{K}_P$ is PSD.

**Decentralized optimization based on one-shot communication:** In this setting, we share the information among all the agents in *one-shot*, then each agent learns its corresponding minimizer using the gathered information. We assume that each agent can communicate with every other agent either in a decentralized manner (or via a central server) before initialization. This is a common assumption in distributed learning with RF kernels where the agents need to share random seeds before initialization to determine the approximate feature mapping (Richards et al., 2020; Xu et al., 2020). Here, consensus is not enforced as each agent can learn a *local* minimizer which has a good *global property*. The label information is also exchanged among all the agents. In Algorithm 2, we list the steps of the algorithm. In Step 3, the agents learn $\mathbf{K}_P$ (the local estimate of the kernel matrix)

using Algorithm 1. In Step 4, the agents share the labels $\bar{y}_m$ so that each agent can (approximately) reconstruct the loss $\hat{\mathcal{L}}(\alpha)$ (cf. (10)) locally. Then each agent can either choose **Option I** or **Option II** to solve (10). A few important properties of Algorithm 2 are:

**[Communication]** Each agent communicates a total of $\mathcal{O}(nP) = \mathcal{O}(NP/M)$ bits (if the norms also need to be transmitted, then with an additional $N/M$ real values) for the GIP kernel, and $\mathcal{O}(NP/M)$ real values for the RF kernels. Importantly, for the GIP kernel the communication is independent of the parameter dimension, making it suitable for decentralized overparameterized learning problems; see Table 1 for a comparison with other approaches.

**[No consensus needed]** Each agent executes Algorithm 2 independently to learn $\alpha_m$, without needing to reach any kind of consensus. They are free to choose different initializations, step-sizes, and even regularizers (i.e., $\lambda$ in (10)). In contrast to the classical learning, where algorithms are designed to guarantee consensus (Koppel et al., 2018; Richards et al., 2020; Xu et al., 2020), our algorithms allow each agent to learn a different function.

The proposed framework relies on sharing matrices $A_m$'s that are random functions of the local features. Note that problem (10) can also be solved by using an iterative distributed gradient tracking algorithm (Qu & Li, 2018), with the benefit that no label sharing is needed; see Appendix D.

*Remark* 4 (Optimization performance). Note that using Algorithm 2, we can solve the approximate problem (10) to arbitrary accuracy using either **Option I** or **Option II**. However, it is by no means clear if the solution obtained by Algorithm 2 will be close to the solution of (3). Therefore, after problem (10) is solved, it is important to understand how close the solutions returned by Algorithm 2 are to the original kernel regression problem (3).

## 4 MAIN RESULTS

In this section, we analyze the performance of Algorithm 2. Specifically, we are interested in understanding the training loss and the generalization error incurred due to the kernel approximation (cf. Algorithm 1). For this purpose, we focus on $\ell_2$ loss functions for which the kernel regression problem (10) can be solved in closed-form. Specifically, we want to minimize the loss:

$$\mathcal{L}(f) = \frac{1}{2}\mathbb{E}_{x,y \sim \pi(x,y)}[(f(x) - y)^2]. \tag{11}$$

We solve the following kernel ridge regression problem with the choice $\hat{\mathcal{L}}(\alpha) = \frac{1}{2N}\|\bar{y} - \mathbf{K}\alpha\|^2$,

$$\min_{\alpha \in \mathbb{R}^N} \left\{ \hat{\mathcal{R}}(\alpha) = \frac{1}{2N}\|\bar{y} - \mathbf{K}\alpha\|^2 + \frac{\lambda}{2}\|\alpha\|_{\mathbf{K}}^2 \right\} \tag{12}$$

where we denote $\bar{y} = [\bar{y}_1^T, \ldots, \bar{y}_M^T]^T \in \mathbb{R}^N$ with $\bar{y}_m = [y_m^{(1)}, y_m^{(2)}, \ldots, y_m^{(n)}]^T \in \mathbb{R}^n$. The above problem can be solved in closed form with $\hat{\alpha}^* = [\mathbf{K} + N \cdot \lambda \cdot I]^{-1}\bar{y}$. The approximated problem at each agent with the kernel $\mathbf{K}_P$ and with the loss function $\hat{\mathcal{L}}_P(\alpha) = \frac{1}{2N}\|\bar{y} - \mathbf{K}_P\alpha\|^2$ is

$$\min_{\alpha \in \mathbb{R}^N} \left\{ \hat{\mathcal{R}}_P(\alpha) = \frac{1}{2N}\|\bar{y} - \mathbf{K}_P\alpha\|^2 + \frac{\lambda}{2}\|\alpha\|_{\mathbf{K}_P}^2 \right\} \tag{13}$$

with the optimal solution returned by **Option I** in Algorithm 2 as $\hat{\alpha}_P^* = [\mathbf{K}_P + N \cdot \lambda \cdot I]^{-1}\bar{y}$. The goal is to analyze the impact of the approximation on the performance of Algorithm 2. Specifically, we bound the difference between the optimal losses of the exact and the approximated Kernel ridge regression. We begin with some assumptions.

**Assumption 1.** We assume $|k(x, x')| \leq \kappa^2$ and $|k_P(x, x')| \leq \kappa^2$ for some $\kappa \geq 1$.

**Assumption 2.** The function $g(\cdot)$ in (5) used to construct the GIP kernel is G-Lipschitz w.r.t. $\psi$, i.e., $\exists G \geq 0$ such that: $|g(\psi, z_2, z_3) - g(\hat{\psi}, z_2, z_3)| \leq G|\psi - \hat{\psi}|, \forall \psi, \hat{\psi} \in [0, \pi]$ and $\forall z_2, z_3 \in \mathbb{R}$.

**Assumption 3.** We assume that the data labels $|y| \leq R$ almost surely for some $R > 0$.

**Assumption 4.** There exists $f_{\mathcal{H}} \in \mathcal{H}$ such that $\mathcal{L}(f_{\mathcal{H}}) = \inf_{h \in \mathcal{H}} \mathcal{L}(h)$.

A few remarks are in order. Note that Assumptions 1, 3 and 4 are standard in the statistical learning theory (Cucker & Zhou, 2007; Caponnetto & De Vito, 2007; Ben-Hur & Weston, 2010; Rudi & Rosasco, 2017). Moreover, for RF kernel Assumption 1 is automatically satisfied if $|\zeta(x, \omega)| \leq \kappa$

almost surely (Rudi & Rosasco, 2017) (cf. (6) and (9)). Assumption 2 is required for estimating the kernel by approximating the pairwise angles between feature vectors. It is easy to verify that the popular kernels including, NTK (15), Arccosine, Gaussian and Polynomial kernels satisfy Assumption 2 with feature vectors belonging to a compact domain (this ensures that the Lipschitz constant $G$ is independent of the feature vector norms). Now we are ready to present the results.

We analyze how well Algorithm 1 approximates the exact kernel. We are interested in the approximation error as a function of the number of random samples $P$. We have the following lemma.

**Lemma 4.1** (Kernel Approximation). *For $\mathbf{K}_P$ returned by Algorithm 1, the following holds with probability at least $1 - \delta$: (i) For the GIP kernel, $\|\mathbf{K} - \mathbf{K}_P\| \leq GN\left(\sqrt{\frac{32\pi^2}{P}\log\frac{2N}{\delta}} + \frac{8\pi}{3P}\log\frac{2N}{\delta}\right)$.*

*(ii) Similarly, for the RF kernel, $\|\mathbf{K} - \mathbf{K}_P\| \leq \kappa^2 N\left(\sqrt{\frac{8}{P}\log\frac{2N}{\delta}} + \frac{4}{3P}\log\frac{2N}{\delta}\right)$.*

Note that as $P$ increases $\mathbf{K}_P \to \mathbf{K}$, in particular, to achieve an approximation error of $\epsilon > 0$, we need $P = \mathcal{O}(\epsilon^{-2})$. Importantly, Lemma 4.1 plays a crucial role in analyzing the optimization performance of the kernel approximation approach. Next, we state the training loss incurred as a consequence of solving the approximate decentralized problem (13) in Algorithm 2 instead of (12).

**Theorem 4.2** (Approximation: Optimal Loss). *Suppose $P \geq \frac{2}{9}\log\frac{2N}{\delta}$, then for both the GIP and the RF kernels, the solution returned by Algorithm 2 (**Option I**) for solving (12) approximately (i.e, (13)), satisfies the following with probability at least $1 - \delta$*

$$\left|\hat{\mathcal{L}}_P(\hat{\alpha}_P^*) - \hat{\mathcal{L}}(\hat{\alpha}^*)\right| = \mathcal{O}\left(\sqrt{\frac{1}{P}\log\frac{2N}{\delta}}\right) \quad and \quad \left|\hat{\mathcal{R}}_P(\hat{\alpha}_P^*) - \hat{\mathcal{R}}(\hat{\alpha}^*)\right| \leq \mathcal{O}\left(\sqrt{\frac{1}{P}\log\frac{2N}{\delta}}\right).$$

Theorem 4.2 states that as $P$ increases, the optimal training loss achieved by solving approximate problem (13) via Algorithm 2 (**Option I**) will approach the performance of the centralized system (12) for both the GIP and the RF kernels. The proof of the above result utilizes Lemma 4.1 and the definition of the loss functions in (12) and (13). See Appendix G for a detailed proof.

The results of Lemma 4.1 and Theorem 4.2 characterize the approximation performance of the proposed approximation – optimization framework on fixed number of training samples. Of course, it is of interest to analyze how the proposed approximation algorithms will perform on unseen test data. Towards this end, it is essential to analyze the performance of the function $\hat{f}_P$ learned from solving (13) via Algorithm 2. We have the following result.

**Theorem 4.3** (Generalization performance). *Let us choose $\lambda = 1/\sqrt{N}$, $\delta \in (0,1)$, and $N \geq \max\left\{\frac{4}{3\|K\|^2}, 72\kappa^2\sqrt{N}\log\frac{32\kappa^2\sqrt{N}}{\delta}\right\}$, also choose $P \geq \max\left\{8, \frac{512\pi^2 G^2}{\|K\|^2}, 288\pi^2 G^2 N\right\}\log\frac{16}{\delta}$ for the GIP kernel and $P \geq \max\left\{8\kappa^2, \frac{32\kappa^2}{\|K\|^2}, 72\kappa^2\sqrt{N}\right\}\log\frac{128\kappa^2\sqrt{N}}{\delta}$ for the RF kernel, where $K$ is defined in Appendix F. Then with probability at least $1 - \delta$, we have for $\hat{f}_P$ returned by Algorithm 2 (**Option I**) for approximately solving (12) (i.e., (13)): $\mathcal{L}(\hat{f}_P) - \inf_{h \in \mathcal{H}}\mathcal{L}(h) = \mathcal{O}(1/\sqrt{N})$.*

The proof of Theorem 4.3 utilizes a result similar to Lemma 4.1 but for integral operator defined using kernels $k(\cdot, \cdot)$ and $k_P(\cdot, \cdot)$. Theorem 4.3 states that with appropriate choice of $\lambda$ (the regularization parameter), $N$ (the number of overall samples), and $P$ (the messages communicated per agent), the proposed algorithm achieves the minimax optimal generalization performance (Caponnetto & De Vito, 2007). Also, note that the the requirement of $P = O(\sqrt{N})$ for the RF kernel compared to $P = O(N)$ for the GIP kernel is due to the particular structure of the RF kernel (cf. (6)). It can be seen from Lemmas H.4 and H.5 in Appendix H, that the approximation obtained with the RF kernel allows the derivation of tighter bounds compared to the GIP kernel. The next corollary precisely states the total communication required per agent to achieve this optimal performance.

**Corollary 1** (Communication requirements for the GIP and RF kernels). *Suppose Algorithm 2 uses the choice of parameters stated in Theorem 4.3 to approximately optimize (12). Then it requires a total of $\mathcal{O}(N^2/M)$ bits (resp. $\mathcal{O}(N\sqrt{N}/M)$ real values) of message exchanges per node when the GIP kernel (resp. the RF kernel) is used, to achieve minimax optimal generalization performance. Moreover, if unnormalized feature vectors are used, then the GIP kernel requires an additional $O(N/M)$ real values of message exchanges per node.*

Compared to DKRR-RF-CM (Liu et al., 2021), Decentralized RF (Richards et al., 2020), DKLA, and COKE (Xu et al., 2020), the number of message exchanges required by the proposed algorithm

Table 2: Total communication (in bits) per node required to achieve a fixed MSE ($\times 10^{-3}$) performance.

| Algorithm | Communication (bits) | | |
|---|---|---|---|
| | $P = 100$ | $P = 500$ | $P = 1000$ |
| DKRR-RF-CM (Liu et al., 2021) | $25,600$ | $640,000$ | $896,000$ |
| DecentralizedRF (Richards et al., 2020) | $57,600$ | $352,000$ | $576,000$ |
| DKLA (Xu et al., 2020) | $44,800$ | $288,000$ | $448,000$ |
| Algorithm 2 (Our Paper) | $22,800$ | $62,800$ | $112,800$ |
| **Target MSE** ($\times 10^{-3}$) | $24.36$ | $20.93$ | $19.25$ |

Table 3: Comparison of MSE for a fixed communication budget.

| Algorithm | MSE ($\times 10^{-3}$) | | |
|---|---|---|---|
| | $P = 100$ | $P = 500$ | $P = 1000$ |
| DKRR-RF-CM (Liu et al., 2021) | $35.30$ | $50.51$ | $67.48$ |
| DecentralizedRF (Richards et al., 2020) | $39.42$ | $43.37$ | $45.77$ |
| DKLA (Xu et al., 2020) | $35.89$ | $43.87$ | $44.73$ |
| Algorithm 2 (Our Paper) | $24.36$ | $20.93$ | $19.25$ |
| **Communication Budget (bits)** | $22,800$ | $62,800$ | $112,800$ |

is independent of the iteration numbers, and it is much less compared to other algorithms, especially for the GIP kernel in the overparameterized regime; see Table 1 for detailed comparisons.

## 5  EXPERIMENTS

We compare the performance of the proposed algorithm to DKRR-RF-CM (Liu et al., 2021), Decentralized RF (Richards et al., 2020), and DKLA (Xu et al., 2020). We evaluate the performance of all the algorithms on real world datasets from the UCI repository.

Specifically, we present the results on National Advisory Committee for Aeronautics (NACA) airfoil noise dataset (Lau & López, 2009), where the goal is to predict aircraft noise based on a few measured attributes. The dataset consists of $N = 1503$ samples that are split equally among $M = 10$ nodes. Each node utilizes $70\%$ of its data for training and $30\%$ for testing purposes. Each feature vector $x_m^{(i)} \in \mathbb{R}^5$ represents the measured attributes such as, frequency, angle, etc., and each label $y_m^{(i)}$ represents the noise level. Additional experiments on different datasets and classification problems, as well as the detailed parameter settings, are included in the Appendix A.

We evaluate the performance of all the algorithms with the Gaussian kernel. Note that the algorithms DKRR-RF-CM, Decentralized RF, and DKLA can only be implemented using the RF approach while our proposed algorithm utilizes the GIP kernel. Also, in contrast to these benchmark algorithms that use iterative parameter exchange, the proposed Algorithm 2 uses only one-shot communication. First, in Table 2, we compare the communication required by each algorithm with the Gaussian kernel for $P = 100, 500$, and $1000$ to achieve the same test mean squared error (MSE) for each setting, see last row of Table 2. Note that for $P = 100$, the communication required by Algorithm 2 is less than $50\%$ of that required by DKLA and Decentralized RF while it is only slightly less than that of DKRR-RF-CM. Moreover, as $P$ increases to $500$ and $1000$, it can be seen that Algorithm 2 only requires a fraction of communication compared to other algorithms, and this fact demonstrates the utility of the proposed algorithms for over-parameterized learning problems. In Table 3, we compare the averaged MSE achievable by different algorithms, when a fixed total communication budget (in bits) is given for each setting (see the last row of Table 3 for the budget). Note that Algorithm 2 significantly outperforms all the other methods as $P$ increases. This is expected since Algorithm 2 essentially solves a centralized problem (cf. Problem (10)) after the multi-agent kernel approximation (cf. Algorithm 1), and a large $P$ provides a better approximation of the kernel (cf. Lemma 4.1). In contrast, for the parameter sharing based algorithms the performance deteriorates even though the kernel approximation improves with large $P$ as learning a high-dimensional parameter naturally requires more communication rounds as well as a higher communication budget per communication round.

Please note that we also compare the performance of Algorithm 2 with the benchmarking algorithms discussed above for the NTK. We further benchmark the performance of Algorithm 2 against the centralized algorithms for the Gaussian, the Polynomial, and the NTK. However, due to space limitations, we relegate these numerical results to the Appendix A.

## ACKNOWLEDGEMENTS

We thank the anonymous reviewers for their valuable comments and suggestions. The work of Prashant Khanduri and Mingyi Hong is supported in part by NSF grant CMMI-1727757, AFOSR grant 19RT0424, ARO grant W911NF-19-1-0247 and Meta research award on "Mathematical modeling and optimization for large-scale distributed systems". The work of Mingyi Hong is also supported by an IBM Faculty Research award. The work of Jia Liu is supported in part by NSF grants CAREER CNS-2110259, CNS-2112471, CNS-2102233, CCF-2110252, ECCS-2140277, and a Google Faculty Research Award. The work of Hoi-To Wai was supported by CUHK Direct Grant #4055113.

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
