# OpenReview forum: "Decentralized Learning for Overparameterized Problems: A Multi-Agent Kernel Approximation Approach"
_ICLR.cc/2022/Conference — ICLR 2022 Poster_

### Official Review · Reviewer_ut1q · 2021-10-23

**Correctness:** 4
**Technical Novelty And Significance:** 3
**Empirical Novelty And Significance:** 2
**Recommendation:** 6
**Confidence:** 4

**Main Review:**

Strengths

1. The paper is clearly written. The contents are well organized and easy to follow.

2. The algorithms and theoretical analyses are technically sound and novel. I read through the algorithms and corresponding discussions. I checked their details and everything looks correct to me. The rkhs setting is new and interesting. The theoretical analyses under the new settings are sound. I read the proof for theorem 4.1 and it looks good to me. These performance and error analyses are classical in the kernel field. Although I didn't read all proofs and there may be mistakes, I think they can be easy to fix.

Weaknesses

1. The experiment and comparison to other methods are weak. As mentioned in the introduction, communicating high-dimensional parameters is problematic and might leading to slow convergence, but I feel this is not convincing because of no sound evidence. I would suggest adding a simple experiment to demonstrate this. It is also unclear how previous approaches perform compared to this work. This work relies on kernels so there can be some shortcomings, e.g. kernels may not be a good choice for large-scale datasets. I would suggest (1) adding experiments to compare this work with previous ones to see when this approach is preferable or to highlight the shortcomings (or advantages) of other methods; (2) adding a table summarizing the difference between this work and others, e.g. whether they use neural network and what are pros and cons.


Minor:

Above Eqn. (5) duplicate ''as''

Below Eqn. (3) '' ... represent (m . i)th element of vector [K \alpha] .. '' redundant to the notation section

Page 5 ''will dependent on the underlying kernel approximation schemes'' -> ''will be dependent ...''




**Summary Of The Paper:**

The decentralization problem has data distributed among many agents and each agent is wanted to maintain some privacy. In this paper, the authors study the decentralized empirical risk minimization problem with reproducing kernel hilbert space. Two large classes of kernels are considered: (1) generalized inner-product (GIP) kernel based on arccosine kernel (proposed in this work), (2) random feature (RF) kernel. In order to attain decentralization, authors approximate kernels based on the inner product of two finite vectors, and propose algorithms (one-shot and iterative) to optimize private models. In addition, authors study in theory the approximation error for kernels, optimization algorithm performance and generalization error. Finally, experiments are presented to validate the algorithms and theoretical results.

**Summary Of The Review:**

Overall, I think this paper is technically sound, has solid theoretical contributions and the paper is clearly written, but it doesn't position itself well in the related area and the experiment is weak. I vote for weak acceptance.

---

> ### Author Response · Authors · 2021-11-17
> **Response: Comparison with the State-of-the-art**
>
> We thank the reviewer for the detailed review of the paper and the encouraging feedback. Below, we address the reviewer's comments in a point-by-point manner.
>
> > **Your comment:** The experiment and comparison to other methods are weak. As mentioned in the introduction, communicating high-dimensional parameters is problematic and might leading to slow convergence, but I feel this is not convincing because of no sound evidence. I would suggest adding a simple experiment to demonstrate this. It is also unclear how previous approaches perform compared to this work. This work relies on kernels so there can be some shortcomings, e.g. kernels may not be a good choice for large-scale datasets. I would suggest (1) adding experiments to compare this work with previous ones to see when this approach is preferable or to highlight the shortcomings (or advantages) of other methods; (2) adding a table summarizing the difference between this work and others, e.g. whether they use neural network and what are pros and cons.
>
>
> **Our Response:**  To address the reviewer's concerns, we have compared the proposed algorithmic framework to the state-of-the-art decentralized kernel learning algorithms. Specifically, we theoretically (please Table 1 below) compare the performance of the proposed algorithm to that of other decentralized kernel learning algorithms. Moreover, in the revised version of the manuscript, we will experimentally verify the performance of the proposed framework to that of the parameter-sharing approaches. We are currently running the experiments and will update the numerical results section shortly with additional experiments.
>
>
> |  Algorithm |  Kernel  |  Bound on $M$  |  Total Messages Communicated |
> |---|---|---|---|
> | DKRR-CM [1]  |  Any |  $O(N^{(T + 1)/2(T + 2)})$ |  $d T N$ |
> |  DKRR-RF-CM [2] |  RF |  $O(N^{(T + 1)/2(T + 2)})$  |  $T P$  |
> |  Decentralized RF [3] | RF  | $O(N^{1/3})$  |  $T P$ |
> |  DKLA/COKE [4]  |  RF | Any $M$  | $T P$  |
> | Ours  |  RF + GIP |  Any $M$ | $N P / M$   |
>
>
> Table 1: Here $N$ is the number of overall samples, $M$ is the number of nodes, $d$ is the data dimension, $P$ is defined as the feature dimension communicated by each node (please see Algorithm 1 in paper). Here $T$ denotes the total communication (iterations) rounds utilized by the distributed algorithm. DKRR-CM [1] relies on sharing data and is therefore not preferred in practical settings. For the RF kernel in the general case (not overparameterized case), the algorithms proposed in our work outperform all the other algorithms for $T > N/M$. Moreover, in the overparameterized regime for the GIP kernel, we only need $P = O(N)$ while for [1-4] we need $P = O(N^{\beta})$ with $\beta > 2$, thereby outperforming all the other algorithms. Finally, note that since our analysis is developed on the multi-agent-kernel-approximation, it does not impose any upper bounds on the number of machines present in the network.
>
> [1] [Lin et al 2020] Shao-Bo Lin, Di Wang, and Ding-Xuan Zhou. Distributed kernel ridge regression with communications. J. Mach. Learn. Res., 21:93–1, 2020
>
> [2] [Liu et al 2021] Yong Liu, Jiankun Liu, and Shuqiang Wang. Effective distributed learning with random features: Improved bounds and algorithms. In International Conference on Learning Representations, 2021.
>
> [3] [Richards et al 2020] Dominic Richards, Patrick Rebeschini, and Lorenzo Rosasco. Decentralised learning with random features and distributed gradient descent. In International Conference on Machine Learning, pp. 8105–8115. PMLR, 2020.
>
> [4] [Xu et al. 2020] Ping Xu, Yue Wang, Xiang Chen, and Tian Zhi. Coke: Communication-censored kernel learning for decentralized non-parametric learning. arXiv preprint arXiv:2001.10133, 2020.
>
> ### Minor comments
>
> > **Your Comment:** Above Eqn. (5) duplicate ''as''
>
> **Our Response:**  We have corrected the typo.
>
> > **Your Comment:** Below Eqn. (3) '' ... represent $(m . i)$th element of vector $[K \alpha]$ .. '' redundant to the notation section
>
> **Our Response:** Thanks. We have removed the redundant explanations.
>
> > **Your Comment:** Page 5 ''will dependent on the underlying kernel approximation schemes'' -> ''will be dependent ...''
>
> **Our Response:**  We have rephrased the sentence.

---

> > ### Comment · Reviewer_ut1q · 2021-11-21
> > **Thanks for your reply**
> >
> > Thank you very much for your detailed replies to all reviews. The reply to mine makes sense. I am looking forward to other reviewers' comments.

---

### Official Review · Reviewer_gN2n · 2021-10-29

**Correctness:** 3
**Technical Novelty And Significance:** 2
**Empirical Novelty And Significance:** 2
**Recommendation:** 5
**Confidence:** 5

**Main Review:**

The main technical contribution in my reading is establishing minimax optimal generalization performance of the proposed technique. My main curiosity comes from whether the strength of this convergence result is through simply applying a stronger analysis technique, or if there is something specific about the derived algorithm that permits this strong result? Put another way, is the generalization performance an artifact of the analysis or a result of a carefully calibrated algorithm? This point is obscured in the current writing.

One of the technical innovations of this work is the use of the generalized inner-product kernel. This subsumes a number of common choices such has polynomial (polynimal ?), Gaussian, Laplace, sigmoid, etc. This is critical to obtaining a degree of privacy preservation as it allows agents to only require sharing knowledge of pairwise angles between feature vectors.

Is sharing the matrix $A_m$ actually more communication efficient than sharing local kernelized function evaluations? This seems like it would actually require more network throughput. Put another way, sharing the parametric representation of each agent's local function is computationally costly, and potentially worse than simply sharing estimates of the local label/target variable. In that way, Algorithm 2 seems much more practical, even it requires label exchange.

In Assumption 2, I am a little concerned about the Lipschitz continuity assumption because of the presence of the norms in the denominator in equation (4). Can we be sure that this is independent of $z_2, z_3$? Some comment about this seems warranted.

In what sense is the convergence theory presented here stronger or sharper than that which is discussed in the introduction for multi-agent optimization over RKHS? Such a granular contrastive discussion is missing from Section 4.

Along the lines of the previous comment, the authors have not compared against any of the other decentralized methods for RKHS optimization experimentally, which makes it difficult to assess whether these results actually sharpen the state of the art in any meaningful way.


Minor comments:

 ``However this approach raises privacy concerns, thus almost never being used in practice." Is awkward syntax. Consider rephrasing.

Also, the main boldface question at the top of page 2 is a sentence fragment.

The statement "where the first one only needs one-shot information exchange, but requires
sharing data labels among the agents; the second one needs iterative information exchange, but does
not need to share the data labels." Is not accurate in the sense that if label exchange is required, then this is mathematically equivalent to data exchange, i.e., realizations of random variables are required... Therefore it is suspicious to state this and immediately afterward state that raw data exchange is not required.


**Summary Of The Paper:**

This work considers multi-agent optimization over RKHS together with random feature approximations. The crux is the development of two different techniques for decentralized computation through a novel introduction of a generalized inner-product kernel. Convergence analysis and numerical validation are provided.

**Summary Of The Review:**

While the paper proposes some interesting methodologies for decentralized optimization over RKHS, it is difficult to assess whether in theory or practice they actually advance the state of the art. This is because they have not done a rigorous comparison of their convergence theory or contrastive numerical experimentation with competing approaches. These aspects need to be thoroughly addressed before I would consider this work to meet the bar for publication.

---

> ### Author Response · Authors · 2021-11-17
> **Response: Contributions and Clarifications**
>
> We thank the reviewer for taking his/her valuable time to provide insightful suggestions for the paper.
>
> > **Your Comment:** The main technical contribution in my reading is establishing minimax optimal generalization performance of the proposed technique. My main curiosity comes from whether the strength of this convergence result is through simply applying a stronger analysis technique, or if there is something specific about the derived algorithm that permits this strong result? Put another way, is the generalization performance an artifact of the analysis or a result of a carefully calibrated algorithm? This point is obscured in the current writing.
>
> **Our Response:** We point out that the major contribution of our work is to develop a decentralized learning framework for overparameterized problems, i.e., very high-dimensional problems.                                                                                                                                                                                                         The main challenge for these problems arises from the fact that parameter sharing incurs a high communication cost as the parameter dimension grows. Our approach relies on approximating the kernel at each node using the novel multi-agent kernel approximation framework proposed in this work that avoids high-dimensional parameter sharing. Please also see the response to “reviewer MEfo” above for a detailed comparison of our framework with state-of-the-art distributed kernel learning methods.
>
> We note that the main challenge in establishing the convergence guarantees for the proposed algorithms is in characterizing the effect of the kernel approximation (for both the GIP and the RF kernels) on the solution of the decentralized kernel regression problem. This is achieved by a careful design of the kernel approximation framework that can be shown to generate approximate kernels which are close to the exact kernel with high probability. For illustration, see Lemmas H4, H5, and H7 in the appendix that capture the effect of the kernel approximation on the generalization performance of the proposed algorithms.
>
> > **Your Comment:** One of the technical innovations of this work is the use of the generalized inner-product kernel. This subsumes a number of common choices such has polynomial (polynomial ?), Gaussian, Laplace, sigmoid, etc. This is critical to obtaining a degree of privacy preservation as it allows agents to only require sharing knowledge of pairwise angles between feature vectors.
>
> **Our Response:** The reviewer is correct in saying that one of the innovations of the work is to identify and propose GIP kernels that are suitable for decentralized implementation. We show that both the GIP and the RF kernels can be efficiently approximated at each node by only sharing some random statistics of the local data. Since the proposed algorithms do not share the complete raw features for implementation and the non-linear transformations, $\zeta$, utilized to transform the data are not one-to-one the proposed algorithms might naturally provide some level of privacy.
>
> > **Your Comment:** Is sharing the matrix $A_m$ actually more communication efficient than sharing local kernelized function evaluations? This seems like it would actually require more network throughput. Put another way, sharing the parametric representation of each agent's local function is computationally costly, and potentially worse than simply sharing estimates of the local label/target variable. In that way, Algorithm 2 seems much more practical, even though it requires label exchange.
>
> **Our Response:** The reviewer is correct in pointing out that sharing the matrix $A_m$ is more communication efficient compared to sharing the local kernel function evaluations. In addition, note that sharing the parametric representation of each agent's local function with other nodes requires sharing raw data (both features and label information) among nodes, since the local functions  $\hat{L}_m(f) = \frac{1}{n} {\sum_i} \ell ( f(x_m^{(i)}) , y_m^{(i)} ) $ depend on the local datasets $ \mathcal{N}_m = \\{ x_m^{(i)}, y_m^{(i)} \\}$ for $i \in \\{1, ..., n \\}$ at each node. We would also like to point out that, we also propose Algorithm 3 (Please see Appendix D) that avoids label sharing among nodes.
>
> > **Your Comment:** In Assumption 2, I am a little concerned about the Lipschitz continuity assumption because of the presence of the norms in the denominator in equation (4). Can we be sure that this is independent of $z_2, z_3$? Some comment about this seems warranted.
>
> **Our Response:** We thank the reviewer for pointing this out. Note that the norms in the denominator of equation (4) will not impact the Lipschitz constant in Assumption 2 because the angle between any two vectors is independent of their norms $z_2$ and $z_3$. We will clarify the above point in the revised version of the manuscript.

---

> > ### Author Response · Authors · 2021-11-17
> > **Response Continued: Contributions and Clarifications**
> >
> > > **Your Comment:** In what sense is the convergence theory presented here stronger or sharper than that which is discussed in the introduction for multi-agent optimization over RKHS? Such a granular contrastive discussion is missing from Section 4.
> >
> > **Our Response:** In the revised version of the paper, we have compared the results of the proposed approach to that of the other decentralized kernel learning algorithms. Below, we pick a few important existing works on distributed kernel learning and list their respective conditions and performance for our considered overparameterized setting. We hope that this will better position our current work.
> >
> > - [Lin et. al., 2020] proposes a distributed kernel learning algorithm that utilizes communicating functional gradients among the nodes at each iteration. Sharing these functional gradients among nodes requires sharing raw data so that these functional gradients can be evaluated (see Steps 1-4 in Section 2.3 of [Lin et. al., 2020]). Therefore, this algorithm is rarely utilized in practical systems where “raw data” sharing is prohibited.
> >
> > - [Liu et al., 2021] focuses on the Random Features (RF) kernel, and proposes a learning algorithm that shares gradients (and Hessian gradient products) of the local losses in each communication round. Since the gradient dimension is the same as the parameter dimension, for high-dimensional overparameterized problems (such as learning with NTK, and when problem dimension >= N^2), sharing these gradients will suffer from high communication costs.
> >
> > - [Richards et al., 2020] and [Xu et al. 2020] focus on RF kernels and propose decentralized kernel learning algorithms that rely on parameter sharing among nodes. Specifically, [Richards et al., 2020] proposes a decentralized gradient descent (DGD), and [Xu et al. 2020] proposes an Alternating Direction Method of Multipliers (ADMM) based algorithm for solving the decentralized kernel learning problem.  Since both the algorithms directly share the high-dimensional parameters they again incur high communication costs compared to the kernel learning framework proposed in our work.
> >
> > Please see Table 1 below for a detailed comparison of these algorithms with our approach. Moreover, note from Table 1 that the algorithms proposed in [Lin et. al., 2020] [Richards et al., 2020] [Liu et al., 2021] all impose an *upper-bound* on the number of nodes $M$. In contrast, our algorithm does not impose any such restriction on the number of nodes. Please see [Figure 2, Lin et. al., 2020] to note the performance deterioration of such algorithms as the number of nodes increases.
> >
> > |   Algorithm | Kernel |   Bound on $M$  |   Total Messages Communicated |
> > |---|---|---|---|
> > |  DKRR-CM [1]  |  Any  |  $O(N^{(T + 1)/2(T + 2)})$  | $d T N$  |
> > |  DKRR-RF-CM [2] |  RF  |  $O(N^{(T + 1)/2(T + 2)})$  |  $T P$ |
> > | Decentralized RF [3]  |  RF  |  $O(N^{1/3})$ | $T P$  |
> > |  DKLA/COKE [4]  |  RF  |  Any $M$  | $T P$  |
> > |  Ours  |   RF + GIP |  Any $M$  |  $NP/M$  |
> >
> > Table 1: Here $N$ is the number of overall samples, $M$ is the number of nodes, $d$ is the data dimension, $P$ is defined as the feature dimension communicated by each node (please see Algorithm 1 in paper). Here $T$ denotes the total communication (iterations) rounds utilized by the distributed algorithm. DKRR-CM [1] relies on sharing data and is therefore not preferred in practical settings. For the RF kernel in the general case (not overparameterized case), the algorithms proposed in our work outperform all the other algorithms for $T > N/M$. Moreover, in the overparameterized regime for the GIP kernel, we only need $P = O(N)$ while for [1-4] we need $P = O(N^{\beta})$ with $\beta > 2$, thereby outperforming all the other algorithms. Finally, note that since our analysis is developed on the multi-agent-kernel-approximation, it does not impose any upper bounds on the number of machines present in the network.
> >
> > [1] [Lin et al 2020] Shao-Bo Lin, Di Wang, and Ding-Xuan Zhou. Distributed kernel ridge regression with communications. J. Mach. Learn. Res., 21:93–1, 2020
> >
> > [2] [Liu et al 2021] Yong Liu, Jiankun Liu, and Shuqiang Wang. Effective distributed learning with random features: Improved bounds and algorithms. In International Conference on Learning Representations, 2021.
> >
> > [3] [Richards et al 2020] Dominic Richards, Patrick Rebeschini, and Lorenzo Rosasco. Decentralised learning with random features and distributed gradient descent. In International Conference on Machine Learning, pp. 8105–8115. PMLR, 2020.
> >
> > [4] [Xu et al. 2020] Ping Xu, Yue Wang, Xiang Chen, and Tian Zhi. Coke: Communication-censored kernel learning for decentralized non-parametric learning. arXiv preprint arXiv:2001.10133, 2020.

---

> > > ### Author Response · Authors · 2021-11-17
> > > **Response Continued: Contributions and Minor Comments**
> > >
> > > > **Your Comment:** Along the lines of the previous comment, the authors have not compared against any of the other decentralized methods for RKHS optimization experimentally, which makes it difficult to assess whether these results actually sharpen the state of the art in any meaningful way.
> > >
> > > **Our Response:**  In addition to Table 1, in the revised version of the manuscript, we will experimentally verify the performance of the proposed framework to that of the parameter-sharing approaches. We are currently running the experiments and will update the numerical results section shortly with additional experiments.  Please also see the response to the previous comment for a detailed comparison of our framework with state-of-the-art distributed kernel learning methods.
> > >
> > > ### Minor comments
> > >
> > > >  **Your Comment:** ``However this approach raises privacy concerns, thus almost never being used in practice." Is awkward syntax. Consider rephrasing.
> > >
> > > **Our Response:**  We have rephrased the sentence in the revised version of the manuscript.
> > >
> > > > **Your Comment:** Also, the main boldface question at the top of page 2 is a sentence fragment.
> > >
> > >  **Our Response:**  We have updated the sentence.
> > >
> > > > **Your Comment:** The statement "where the first one only needs one-shot information exchange, but requires sharing data labels among the agents; the second one needs iterative information exchange, but does not need to share the data labels." Is not accurate in the sense that if label exchange is required, then this is mathematically equivalent to data exchange, i.e., realizations of random variables are required... Therefore it is suspicious to state this and immediately afterward state that raw data exchange is not required.
> > >
> > > **Our Response:**  We thank the reviewer for pointing this out. By raw data exchange we imply that the "complete" data including the labels and the features, i.e., $\\{x_m^{(i)}, y_m^{(i)}\\}_{i=1}^n$ is being shared. As noted by the reviewer, to implement the one-shot algorithm we only share label information, however, we do not require the features to be shared. Therefore, we state that the raw data is not exchanged. We also note that label sharing can be avoided by utilizing an iterative algorithm (cf. Algorithm 3 in the Appendix). In the revised version of the manuscript, we have clarified this aspect of the proposed framework.

---

### Official Review · Reviewer_MEfo · 2021-10-30

**Correctness:** 3
**Technical Novelty And Significance:** 2
**Empirical Novelty And Significance:** Not applicable
**Recommendation:** 6
**Confidence:** 4

**Main Review:**

I find it difficult to position this paper in the existing literature on distributed kernel learning.

For example, if the focus is on privacy issues and on not exchanging local data or labels, then more on privacy considerations should be discussed, e.g., in Algorithm 2 it is still required to exchange labels, the authors mention on page 7 that there are ways to avoid such privacy leakage, but details are deferred to the appendix. If the major contribution is the proposed method needs less communication, then it is then *necessary* to compare *explicitly* in which case/regime the proposed method improves previous results such as [Liu et al., 2021]. For the moment, it is not clear, at least to me as a reader, to which extend the results are significant.

I would consider changing my scores if the authors could clarify the major contribution in this paper.

----

**After rebuttal**: I thank the authors for their clarification and their efforts in updating the paper. I believe the contribution of this paper is now much clearer and I've updated my score accordingly.

**Summary Of The Paper:**

This paper discusses a random feature-based multi-agent kernel learning approach. For both generalized inner-product (GIP) and random feature (RF) kernels, the authors propose, in each agent, to exchange the random feature matrix (instead of the model parameters). By considering the problem of kernel ridge regression, some theoretical results including the kernel matrix approximation error (Lemma 4.1), training (Theorem 4.2), and generalization performance (Theorem 4.3) are obtained in Section 4. Some numerical experiments on UCI datasets are provided in Section 5.

The authors argue (e.g., in Corollary 1) that the proposed approach is more efficient as it requires less communication to achieve min-max optimal generalization performance.

**Summary Of The Review:**

Some detailed comments:

* page 5 [Approximation for GIP kernel]: I get confused here because the authors first assume the feature vectors are normalized (which is OK but makes the GIP kernel defined in Equation (4) less interesting, which, in essence, only depends on the "angle" $\psi(x,x')$) and then further takes the nonlinear function $\zeta$ to a binary function: is there any evidence or previous results saying what type of kernel can one approximation with this particular choice of nonlinearity? At least, this is a much less general family of kernels than the GIP kernel defined in Equation (4).
* the theoretical results in Lemma 4.1, Theorem 4.2, and 4.3 are interesting: while it is stated in Lemma 4.1 that one needs at least $P = O(N)$ random features to well approximate the (whole) kernel matrix up to some $O(1)$ error, it turns out, in Theorem 4.2 that one only needs much less ($P = O(\log(N))$) to get similar training error in the distributed setting. Also, while the results for GIP and RF kernel in Lemma 4.1and Theorem 4.2 are somewhat similar (up to constants), the results in Theorem 4.3 for these two kernels are very different and one needs only $P = O(\sqrt{N})$ for random features but $P = O(n)$ for GIP kernel. Could the authors comment, or perhaps also provide some intuition on this (which may be of independent interest)?

---

> ### Author Response · Authors · 2021-11-16
> **Response: Contributions**
>
> > **Your comment:** I find it difficult to position this paper in the existing literature on distributed kernel learning.
>
> >
> > For example, if the focus is on privacy issues and on not exchanging local data or labels, then more on privacy considerations should be discussed, e.g., in Algorithm 2 it is still required to exchange labels, the authors mention on page 7 that there are ways to avoid such privacy leakage, but details are deferred to the appendix. If the major contribution is the proposed method needs less communication, then it is then necessary to compare explicitly in which case/regime the proposed method improves previous results such as [Liu et al., 2021]. For the moment, it is not clear, at least to me as a reader, to which extent the results are significant.
>
> >
> >I would consider changing my scores if the authors could clarify the major contribution in this paper.
>
>
>
> **Our response:** We thank the reviewer for devoting valuable time to provide insightful comments and suggestions for the paper. Below, we address the reviewer's concerns.
>
> We would like to clarify that the overarching goal of our work is to develop a *communication efficient* decentralized learning framework for overparameterized problems.  The main challenge arises from the fact that direct parameter sharing incurs high communication costs for the class of over-parameterized problems. Below, we pick a few important existing works on distributed kernel learning and list their respective conditions and performance for our considered overparameterized setting. We hope that this will better position our current work.
>
> - [Lin et. al., 2020] proposes a distributed kernel learning algorithm that utilizes communicating functional gradients among the nodes at each iteration. Sharing these functional gradients among nodes requires sharing raw data so that these functional gradients can be evaluated (see Steps 1-4 in Section 2.3 of [Lin et. al., 2020]). Therefore, this algorithm is rarely utilized in practical systems where “raw data” sharing is prohibited.
>
> - [Liu et al., 2021] focuses on the Random Features (RF) kernel and proposes a learning algorithm that shares gradients (and Hessian gradient products) of the local losses in each communication round. Since the gradient dimension is the same as the parameter dimension, for high-dimensional overparameterized problems (such as learning with NTK, and when problem dimension $>= N^2$), sharing these gradients will suffer from high communication costs.
>
> - [Richards et al., 2020] and [Xu et al. 2020] focus on RF kernels and propose decentralized kernel learning algorithms that rely on parameter sharing among nodes. Specifically, [Richards et al., 2020] proposes a decentralized gradient descent (DGD), and [Xu et al. 2020] proposes an Alternating Direction Method of Multipliers (ADMM) based algorithm for solving the decentralized kernel learning problem.  Since both the algorithms directly share the high-dimensional parameters they again incur high communication costs compared to the kernel learning framework proposed in our work.
>
> Please see Table 1 below in this response for a detailed comparison of these algorithms with our approach. Moreover, note from Table 1 that the algorithms proposed in [Lin et. al., 2020] [Richards et al., 2020] [Liu et al., 2021] all impose an *upper-bound* on the number of nodes $M$. In contrast, our algorithm does not impose any such restriction on the number of nodes. Please see [Figure 2, Lin et. al., 2020] to note the performance deterioration of such algorithms as the number of nodes increases.
>
>
> | **Algorithm**      | **Kernel** | **Bound on $M$** | **Total Messages Communicated** |
> |---|---|---|---|
> |DKRR-CM [1] | Any      | $O(N^{(T + 1)/2(T + 2)})$  | $d T N$ |
> | DKRR-RF-CM [2] |  RF | $O(N^{(T + 1)/2(T + 2)})$ |  $T P$ |
> | Decentralized RF [3] | RF |  $O(N^{1/3})$ | $T P$ |
> | DKLA/COKE [4] | RF | Any $M$ | $T P$ |
> | Ours | RF + GIP | Any $M$ | $N P / M$ |
>
> Table 1: Here $N$ is the number of overall samples, $M$ is the number of nodes, $d$ is the data dimension, $P$ is defined as the feature dimension communicated by each node (please see Algorithm 1 in paper). Here $T$ denotes the total communication (iterations) rounds utilized by the distributed algorithm. DKRR-CM [1] relies on sharing data and is therefore not preferred in practical settings. For the RF kernel in the general case (not overparameterized case), the algorithms proposed in our work outperform all the other algorithms for $T > N/M$. Moreover, in the overparameterized regime for the GIP kernel, we only need $P = O(N)$ while for [1-4] we need $P = O(N^{\beta})$ with $\beta > 2$, thereby outperforming all the other algorithms. Finally, note that since our analysis is developed on the multi-agent kernel approximation, it does not impose any upper bounds on the number of machines present in the network.

---

> > ### Author Response · Authors · 2021-11-16
> > **Response Continued: Contributions**
> >
> > **Our response (continued):** From the above discussion, it is desirable to develop algorithms that *do not* share high-dimensional parameters (or gradients) for learning. Therefore, the specific focus of our work is *not* to develop privacy-preserving algorithms, however, we want the developed algorithms to communicate in some *low-dimensional space* for the overparameterized problems, while avoiding raw data (or features) sharing among local nodes.  Also please note that, although Algorithm 2 in the paper does require sharing the label, Algorithm 3 (in the appendix) does not. Importantly, we also note that the proposed algorithm might naturally provide some level of privacy since the non-linear transformations, $\zeta$, utilized to transform the data are not one-to-one transformations. Indeed, developing privacy-preserving algorithms for our decentralized kernel learning framework is an interesting future direction. One possible way is that we can add noise to the messages exchanged between the agents in Algorithm 3 to make the algorithm differentially private. In this case, please note that working on the lower-dimensional space can potentially reduce the total amount of noise added to the messages (compared with, say, directly adding noises to the original high-dimensional parameter space). But due to the space limitation of this work, we will defer this line of investigation to a follow-up paper. We thank the reviewer for pointing this out.
> >
> > To be more specific, in this work we develop a decentralized kernel learning framework that departs from the classical parameter-sharing based learning approaches. Our approach relies on approximating the optimization problem at each node using the multi-agent kernel approximation framework, thus avoiding high-dimensional parameter sharing. We focus on two types of kernels, Generalized Inner-Product (GIP) and Random Features (RF) kernels, whose kernel matrices can be efficiently estimated at each node while sharing only some random statistics of the local data. The key is to exploit the structure in constructing these kernel matrices. Importantly, we show that the proposed learning framework provides state-of-the-art performance guarantees without sharing any "high-dimensional parameters" or "raw data" among nodes.
> >
> > [1] [Lin et al 2020] Distributed kernel ridge regression with communications. JMLR, 2020
> >
> > [2] [Liu et al 2021] Effective distributed learning with random features: Improved bounds and algorithms. ICLR 2021
> >
> > [3] [Richards et al 2020] Decentralised learning with random features and distributed gradient descent. ICML, 2020
> >
> > [4] [Xu et al. 2020] Coke: Communication-censored kernel learning for decentralized non-parametric learning. arXiv 2020

---

> > > ### Author Response · Authors · 2021-11-16
> > > **Response to Detailed Comments**
> > >
> > > > **Your Comment:** page 5 [Approximation for GIP kernel]: I get confused here because the authors first assume the feature vectors are normalized (which is OK but makes the GIP kernel defined in Equation (4) less interesting, which, in essence, only depends on the "angle" $\psi(x, x')$) and then further takes the nonlinear function $\zeta$ to a binary function: is there any evidence or previous results saying what type of kernel can one approximate with this particular choice of nonlinearity? At least, this is a much less general family of kernels than the GIP kernel defined in Equation (4).
> > >
> > > **Our Response:** We thank the reviewer for raising this concern. We note that when the feature vectors are not normalized, the GIP kernel can easily be implemented by only communicating additional $n$ real values of the norms of the local feature vectors. Based on the reviewer's suggestion, we have amended Algorithm 2 and the GIP kernel evaluation to account for the unnormalized feature vectors. We also note that for many applications of interest, as pointed out in (Bietti & Mairal, 2019; Geifman et al., 2020; Pedregosa et al., 2011), normalized feature vectors are utilized for learning, therefore, communicating the norms of the feature vectors can be avoided for such problems.
> > >
> > > Moreover, as highlighted by the reviewer, we choose non-linearity $\zeta$ to estimate the GIP kernel as it maps the feature vectors to binary-valued vectors. We choose this particular non-linearity since it generates mappings that are suitable for communication. To the best of our knowledge, our work is the first that chooses $\zeta$ as a binary mapping to approximate the GIP kernel.
> > >
> > > > **Your Comment:** the theoretical results in Lemma 4.1, Theorem 4.2, and 4.3 are interesting: while it is stated in Lemma 4.1 that one needs at least $P = O(N)$ random features to well approximate the (whole) kernel matrix up to some $O(1)$ error, it turns out, in Theorem 4.2 that one only needs much less $(P = O(\log(N)))$ to get similar training error in the distributed setting. Also, while the results for GIP and RF kernel in Lemma 4.1 and Theorem 4.2 are somewhat similar (up to constants), the results in Theorem 4.3 for these two kernels are very different and one needs only $P = O(\sqrt{N})$ for random features but $P = O(N)$ for GIP kernel. Could the authors comment, or perhaps also provide some intuition on this (which may be of independent interest)?
> > >
> > > **Our Response:** We thank the reviewer for the suggestion. In the revised version of the paper, we have added additional discussions with the theoretical results to address the reviewer's concerns.
> > >
> > > First, note that Lemma 4.1 bounds the approximation error for the estimated kernel $\\|\mathbf{K} - \mathbf{K}_P\\|$, while Theorem 4.2 bounds the approximation error for the optimization problem $ | \hat{L}_P ( \hat{\alpha}^*_P) - \hat{L}(\hat{\alpha}^*) | $. Therefore, the requirement on P is not the same as the approximation errors are evaluated using different measures. We would also like to clarify that the disparity in the requirement for the number of features $P$ in Lemma 4.1 and Theorem 4.2 arises from the definition of the loss functions $\hat{\mathcal{L}}(\alpha)$ and $\hat{\mathcal{L}}_P(\alpha)$. Specifically, note that the loss functions defined as $\hat{\mathcal{L}}(\alpha) = \frac{1}{2N} \\| \bar{y} - \mathbf{K} \alpha \\|^2$ and $\hat{\mathcal{L}}_P(\alpha) = \frac{1}{2N} \\| \bar{y} - \mathbf{K}_P \alpha \\|^2$ are the empirical averages of the individual losses computed at each data point. Note that compared to Lemma 4.1, the presence of factor 1/N in the loss functions of Theorem 4.2 creates the disparity on the requirement of $P$. The details can be seen in the proof of Theorem G.5 in the appendix.
> > >
> > > Moreover, the requirement of $P = O(\sqrt{N})$ for the RF kernel compared to $P = O(N)$ for the GIP kernel is due to the particular structure of the RF kernel. It can be seen from Lemmas H4 and H5 in the appendix, that the approximation obtained with the RF kernel allows the derivation of tighter bounds compared to the GIP kernel.

---

### Author Response · Authors · 2021-11-21
**General response to the reviewers**

**General response to the reviewers.** We thank the reviewers for taking their valuable time to provide insightful comments and suggestions for the paper. We believe that the thoughtful reviews and the recommendations made by the reviewers have substantially improved the quality of the paper. Based on the reviewers' suggestions, we have updated the paper and have highlighted the changes in the blue-colored font. Also to address some of the reviewers’ concerns, we have conducted additional numerical experiments to compare the performance of our approximation -- optimization framework to the state-of-the-art decentralized kernel learning algorithms. In this response, we highlight the major changes to the manuscript.

- To highlight the advantages of our approach, we have updated the ”Introduction” section with a comparison of our approach to the state-of-the-art decentralized learning algorithms for both the overparameterized and the non overparameterized regimes. Please see Table 1 and the discussion in the Introduction section.
- For a better understanding of the decentralized kernel learning problem in the overparameterized regime, we have updated the “Problem Statement” section with a discussion of the overparameterized regime.
- Based on Reviewer MEfo’s suggestion, we have updated Algorithm 1 and the discussion in Section 3 to account for the case when the feature vectors will not be normalized.
- In Section 4, we have added additional explanations and discussion with the assumptions and the theoretical results to address the reviewers’ concerns.
- Finally, to address the reviewers’ major concern of a lack of comparison with the state-of-the-art algorithms, we have conducted additional numerical experiments on multiple UCI datasets to evaluate the performance of the proposed algorithm against the benchmarking algorithms. Please see Section 5 and Appendix A for the comparison of the algorithms.

---

### Decision · Program_Chairs · 2022-01-20

**Decision:**

Accept (Poster)

**Comment:**

In this paper, the authors study the decentralized empirical risk minimization problem with Reproducing Kernel Hilbert Space. I found the problem formulation and the solution quite interesting. The authors also answered the main comments of the reviewers. Even though part of the work is incremental, I feel that there is enough merit to accept this paper.